

# Assessing soil bacterial community and dynamics by integrated high-throughput absolute abundance quantification

Jun Lou[1,*], Li Yang[1,*], Haizhen Wang[1], Laosheng Wu[1,2] and Jianming Xu[1]

[1] Institute of Soil and Water Resources and Environmental Science, College of Environmental and Resource Sciences, Zhejiang Provincial Key Laboratory of Agricultural Resources and Environment, Zhejiang University, Hangzhou, China
[2] Department of Environmental Sciences, University of California, Riverside, United States of America
[*] These authors contributed equally to this work.

Corresponding authors
Haizhen Wang, wanghz@zju.edu.cn
Jianming Xu, jmxu@zju.edu.cn

## ABSTRACT

Microbial ecological studies have been remarkably promoted by the high-throughput sequencing approach with explosive information of taxonomy and relative abundance. However, relative abundance does not reflect the quantity of the microbial community and the inter-sample differences among taxa. In this study, we refined and applied an integrated high-throughput absolute abundance quantification (iHAAQ) method to better characterize soil quantitative bacterial community through combining the relative abundance (by high-throughput sequencing) and total bacterial quantities (by quantitative PCR). The proposed iHAAQ method was validated by an internal reference strain EDL933 and a laboratory strain WG5. Application of the iHAAQ method to a soil phenanthrene biodegradation study showed that for some bacterial taxa, the changes of relative and absolute abundances were coincident, while for others the changes were opposite. With the addition of a microbial activity inhibitor ($NaN_3$), the absolute abundances of soil bacterial taxa, including several dominant genera of *Bacillus*, *Flavobacterium*, and *Paenibacillus*, decreased significantly, but their relative abundances increased after 28 days of incubation. We conclude that the iHAAQ method can offer more comprehensive information to reflect the dynamics of soil bacterial community with both relative and absolute abundances than the relative abundance from high-throughput sequencing alone.

## INTRODUCTION

Bacteria are the essential components of soil ecosystem and play a vital role in the material cycle and energy flow in the Earth ecosystem (*Whitman, Coleman & Wiebe, 1998*). It was estimated that there are $4$–$6 \times 10^{30}$ cells of prokaryotes on Earth and $2.6 \times 10^{30}$ in soil, which is the most diverse and abundant cellular life form on Earth (*Whitman, Coleman & Wiebe, 1998*; *Curtis & Sloan, 2005*). Soil bacteria have been studied intensively for more than a hundred years from individuals and its communities, but there are still more than 85–99% of bacteria that cannot be isolated and grown in culture mediums (*Lok, 2015*). Therefore,

the question remains how many bacteria are there (absolute and relative abundances) and who are they (kinds of species) in a soil, which consists a community.

Over the past few decades, scientists have attempted to answer these questions. The classical plate-counting method via counting the different colonies on plate was the simplest way to simultaneously obtain the information of absolute and relative abundances of soil bacteria, but the method has the limitation because of the low percentage of cultivable bacteria in soil (*Davey, 2011*; *Blagodatskaya & Kuzyakov, 2013*). Many other methods have been applied to explore the absolute or relative abundances of soil bacterial community. For example, *Brookes et al. (1985)* and *Vance, Brookes & Jenkinson (1987)* tried to estimate the total microbial abundance by measuring the microbial biomass-N (MBN) and -C (MBC) in soil. With the development of molecular biology and the more verified specific target genes, the quantitative PCR (qPCR) also becomes a powerful and accurate technique to measure the total or specific microbial absolute abundance in soil (*Philippot et al., 2009*).

Various biomarker and molecule methods were also found to be useful for exploring the community sturctures of soil bacteria. These methods include the phospholipid fatty acids (PLFAs), polymerase chain reaction-denaturing gradient gel electrophoresis (PCR-DGGE), clone library, catalysed reporter deposition fluorescence *in situ* hybridization (CARD-FISH), microfluidic qPCR, etc. (*Li et al., 2006*; *Ding et al., 2011*; *Cai et al., 2016*; *Kleyer, Tecon & Or, 2017*). In particular, establishment of the high-throughput sequencing method opened a new era for microbiology. Confirmed by the microbial mock community and multiple studies, this method can reveal the relative abundance as well as kinds of species of soil bacterial community with unprecedented amount of information (*Wang et al., 2007*; *Caporaso et al., 2012*; *Cai et al., 2016*; *Singer et al., 2016*; *Tourlousse et al., 2017*).

Since the high-throughput sequencing technique can acquire rich information of relative abundance (*Caporaso et al., 2012*; *Kozich et al., 2013*; *Dannemiller et al., 2014*; *Prest et al., 2014*; *Stokell, Hamp & Steck, 2016*; *Props et al., 2017*; *Zhang et al., 2017*), it was combined with the absolute quantification methods of total microorganism to characterize the more informative changes of microbial communities in several studies (*Dannemiller et al., 2014*; *Prest et al., 2014*; *Props et al., 2017*; *Zhang et al., 2017*). In the combined methods, the flow cytometric (FCM), adenosine triphosphate (ATP), heterotrophic plate count (HPC), qPCR, PLFAs, and MBC, were used to quantify the absolute abundance of total microorganism. For example, *Prest et al. (2014)* combined the FCM and 16S rRNA gene pyrosequencing and successfully revealed the change of bacterial community in drinking water, which was undetectable by the ATP and HPC measurements. *Dannemiller et al. (2014)* quantitatively compared the fungal aerosol populations by combining the qPCR and high-throughput sequencing methods, which were verified by the four selected specific taxa, *Alternaria alternata*, *Cladosporium cladosporioides*, *Epicoccum nigrum*, and *Penicillium/Aspergillus* spp.

Recently, *Zhang et al. (2017)* investigated soil bacterial communities at two different locations by coupling the results of ATP, qPCR, PLFAs, and the relative abundances of high-throughput sequencing technique. Nevertheless, the methods were not verified using specific strains or internal strains as it was done in the previous study by *Dannemiller et al. (2014)*. So far, no research has proved the feasibility of combining the absolute

quantification methods of total microorganism and high-throughput sequencing data to characterize soil bacterial communities, which is much more complex than in other environments (*Whitman, Coleman & Wiebe, 1998*; *Smets et al., 2016*; *Tourlousse et al., 2017*). As mentioned above, the qPCR method is an accurate method for the absolute abundance of genes, while the high-throughput sequencing is the primary method for the relative abundance of bacteria community. Therefore, we proposed an integrated high-throughput absolute abundance quantification (iHAAQ) method for soil quantitative bacterial ecology through combining the qPCR and high-throughput sequencing techniques in this study.

Meanwhile, the accuracy of the iHAAQ method was tested by an internal reference strain, *Escherichia coli* O157:H7 strain EDL933, which can be easily classified and distinguished from soil indigenous bacteria. Also, the internal reference strain can be absolutely quantified by the qPCR method because it contains the flagellar biosynthesis protein FliC gene (*fliC*) (*Latif et al., 2014*). Furthermore, the iHAAQ method was applied to characterize the bacterial community changes in soil phenanthrene biodegradation.

## MATERIALS AND METHODS

### Bacteria strains, culture medium and chemicals

The *Escherichia coli* O157:H7 strain EDL933 (ATCC 43895), SMAC (Sorbitol MacConkey)-BCIG (5-bromo-4-chloro-3-indoxyl-β-D-glucuronide) agar (LabM, Lancashire, UK) and Luria-Bertani (LB) medium (Becton, Dickinson and Company, Franklin Lakes, NJ, USA) used herein were described previously (*Wang et al., 2014*). After incubating at 37 °C on a water orbital shaker (250 rpm), the cultured strain EDL933 was harvested by centrifugation at 4 °C (6,000 × g for 10 min) and washed three times with sterile deionized water (*Wang et al., 2014*). The cell pellets were re-suspended in sterile deionized water to achieve the cell concentration of near $2.1 \times 10^{11}$ CFU mL$^{-1}$ (CFU, colony-forming units). Diluted with sterile deionized water, the gradient concentrations of strain EDL933 in the order of $2.1 \times 10^{11}$ to $2.1 \times 10^{6}$ CFU mL$^{-1}$ were prepared and added to soil samples.

The *Massilia* sp. strain WG5 (CCTCC AB 2015362) was isolated from a PAHs contaminated soil from Jiangsu Province, China, which has a high phenanthrene-degrading ability (*Lou et al., 2016*; *Wang et al., 2016*). After pre-cultured in LB medium, the strain WG5 was harvested by centrifugation at 4 °C (6,000 × g for 10 min) and washed three times with sterile deionized water. The cell pellets were then re-suspended to $D_{600nm}$ 1.0 (approximately $1.5 \times 10^{8}$ CFU mL$^{-1}$) for later use.

Phenanthrene (PHE; Sigma-Aldrich, Shanghai, China), rifampicin (Thermo Fisher Scientific, Waltham, MA, USA), nalidixic acid (Sigma-Aldrich, St. Louis, MO, USA) and other inorganic chemicals used in this study were of the HPLC or analytical grade. The extraction and detection of PHE in soil were followed by the method of *Gu et al. (2017)*.

### Experiment set-up and DNA extractions

The soil was collected from the surface horizon (0–20 cm) near a steel mill in Fujian Province, China. Three composite soil samples were taken to the laboratory using coolers

with ice bags. Then, the soil was sieved (2 mm) and stored at 4 °C for use. The soil chemical and physical properties are the same as reported by *Gu et al. (2017)*.

The experiment included 10 treatments: control (Treatment Cont, soil only), soil added with strain EDL933 at six concentrations (near $10^9$, $10^8$, $10^7$, $10^6$, $10^5$, and $10^4$ CFU (g dry wt soil)$^{-1}$, corresponding to Treatments E9, E8, E7, E6, E5 and E4), soil added with PHE (100 $\mu$g (g dry wt soil)$^{-1}$, Treatment P), soil added with PHE (100 $\mu$g (g dry wt soil)$^{-1}$) and strain WG5 (approximately $1.00 \times 10^7$ CFU (g dry wt soil)$^{-1}$, Treatment W), and soil added with PHE (100 $\mu$g (g dry wt soil)$^{-1}$) and sodium azide (NaN$_3$, 0.1%, w/w, Treatment S). The purpose of adding NaN$_3$ was to inhibit the native microbial activity (*Wei & Pan, 2010*).

In Treatments E9, E8, E7, E6, E5 and E4, 1.0-mL aliquot of strain EDL933 with the gradient concentrations in the order of $2.1 \times 10^{11}$ to $2.1 \times 10^6$ CFU mL$^{-1}$, respectively, was added to each soil sample (30 g frozen-dry basis). Treatments E9, E8, E7, E6, E5, E4, and Cont were used to develop the iHAAQ method, and Treatments P, W, and S were used to reveal soil bacterial community dynamics during PHE biodegradation.

Samples of each treatment were prepared in triplicate by putting 30 g (frozen-dry basis) of soil into a 100-mL flask. All the treated soils were thoroughly mixed (two hours) and soil moisture was adjusted to 100% of the field capacity (soil water content at $-33$ kPa) by adding sterile deionized water. Ten grams (frozen-dry basis) of soil were sampled immediately from each flask. In addition, all flasks of Treatments P, W, and S were incubated in the dark at 28 $\pm$ 1 °C and aerated weekly on the Clean Bench for 30 min. Sterile deionized water was added to compensate the soil moisture loss during incubation and to maintain quasi-constant soil moisture content for the samples. After 28 days of incubation, ten grams (frozen-dry basis) of soil were sampled from each flask of Treatments P, W, and S for DNA extraction.

All the soil samples collected from the flasks were immediately frozen and stored in a freezer ($-80$ °C). Aliquot sample of 0.21 g (frozen-dry basis) per collected sample was used for DNA extraction using the MoBio PowerSoil DNA Isolation kit (MoBio Laboratories, Carlsbad, CA, USA). The extracted DNA was stored in the same freezer ($-80$ °C) before qPCR analysis.

## Quantitative PCR analyses

The qPCR analyses were performed to quantify the specific genes of the extracted DNA with three replicates using a StepOnePlus$^{TM}$ Real-Time PCR System instrument (Applied Biosystems, Foster City, CA, USA). The two pairs of primers, 341F (5′-CCTACGGGAGGCAGCAG-3′) and 534R (5′-ATTACCGCGGCTGCTGG-3′), 520F (5′-AYTGGGYDTAAAGNG-3′) and 802R (5′-TACNVGGGTATCTAATCC-3′), were chosen to detect the copy numbers of 16S rRNA gene of total bacteria in V3 and V4 variable regions, respectively (*Reddy et al., 2012*; *Guinane et al., 2013*). To measure the abundance of strain EDL933, the *fliC* gene was selected from its chromosome, which had only one copy in the genome (*Latif et al., 2014*). The forward and reverse primers used were 5′-GCGAAGTTAAACACCACGAC-3′ and 5′-ACCCGTACCAGCAGTAGATT-3′, respectively (*Jun et al., 2011*).

The standard curves were generated using 10-fold serial dilution of a plasmid containing the targeted DNA PCR fragments of 16S rRNA and the *fliC* genes. The 25 µL qPCR reactions contained 12.5 µL 2 × SYBR Green qPCR SuperMix-UDG with Rox (Invitrogen, Carlsbad, CA, USA), 0.5 µL of each 10-µM forward and reverse primers, and 10.5 µL sterile and DNA-free water. The known amount and soil DNA samples were added at 1.0 µL per reaction. The reaction was executed under the following thermocycler conditions: 50 °C for 2 min × 1 cycle, 95 °C for 5 min × 1 cycle, then 95 °C for 15 s, and 60 °C for 30 s × 40 cycles; and finally followed by the dissociation stage for the melting curve analysis. All the gene copy numbers were calculated from the standard curves of 16S rRNA and the *fliC* genes by using the $\Delta C_t$ (cycle threshold) method. All qPCR reactions were conducted in triplicate.

## High-throughput sequencing and data processing

According to the method reported by *Ma et al. (2016)*, each composite DNA sample for sequencing was mixed by combining equal amounts of DNA extraction of the three replicates. The bacterial communities of the Cont, E9 to E4, P, W, and S treatments were amplified with the V4 region of primers (520F, 5′-AYTGGGYDTAAAGNG-3′ and 802R, 5′-TACNVGGGTATCTAATCC-3′), and then sequenced with Illumina Miseq platform following standard protocols. The quality control of raw sequencing reads was performed using QIIME software (version 1.7.0) (*Caporaso et al., 2010*). The sequences were removed if the average quality in a moving 5 bps window lower than Q20, length less than 150 bps and having the ambiguous barcode. The paired-end sequences were spliced by the same index with overlap no less than 10 bps and no mismatch using FLASH (*Magoc & Salzberg, 2011*). The chimera sequences were *de novo* identified and removed using UCHIME (*Edgar et al., 2011*).

After quality filtering, the sequences clustering was performed using UCLUST with a 97% similarity and sorted out as operational taxonomic units (OTUs) (*Edgar, 2010*). Then, the OTUs were classified using RDP-classifier to match to SILVA database release 119 (*Wang et al., 2007*; *Quast et al., 2013*). The raw data obtained in this research were deposited to NCBI SRA (Sequence Read Archive; http://www.ncbi.nlm.nih.gov/sra/) under accession numbers SRP097773 and SRP105351.

## Absolute quantification of soil bacteria and its community by the iHAAQ method

The absolute abundance of each phylum or genus in the soil bacterial community was calculated by multiplying total bacterial quantities (the copy number of 16S rRNA gene of total bacteria from qPCR) and the corresponding relative abundance from high-throughput sequencing (Fig. 1). The calculated absolute abundances of *Escherichia-Shigella*, Esch-V3 and Esch-V4, were based on its relative abundance and the total bacterial quantities measured by V3 and V4 regions of 16S rRNA gene, respectively. Since the 16S rRNA gene varied in copy numbers among the different bacteria genomes (*Latif et al., 2014*; *Sun et al., 2013*), seven copies of 16S rRNA gene and 1 copy of *fliC* gene in one genome of strain EDL993 (NZ_CP008957, NCBI: http://www.ncbi.nlm.nih.gov) were considered in this

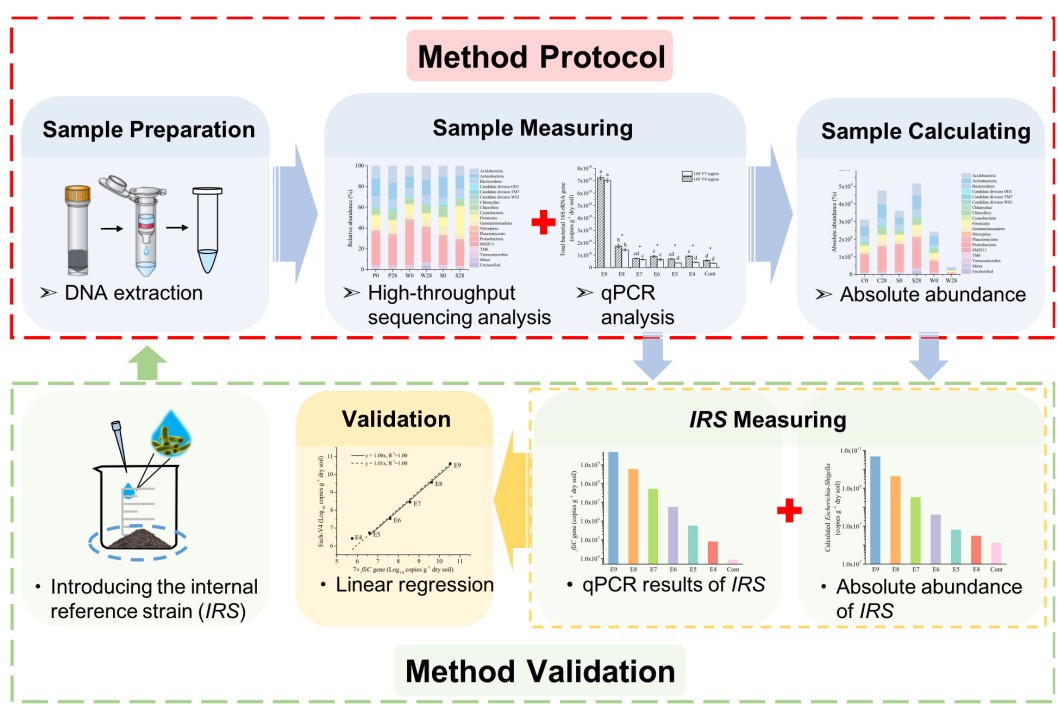

**Figure 1** **A schematic workflow of the iHAAQ method and its validation procedure.** For method validation, the internal reference strain (*IRS*) was introduced into soil before DNA extraction and measured by iHAAQ and qPCR methods. These two results were combined and analyzed by liner regression to validate the iHAAQ method.

study. The absolute abundances of *Escherichia-Shigella* (Esch-V3 and Esch-V4) and $7\times$ *fliC* gene copies were logarithmically transformed before performing linear regression analysis.

Linear regression was performed using Origin 2016 (OriginLab, Northampton, MA, USA), and analysis of variance (ANOVA) used with SPSS 20 (IBM, Armonk, USA). The homogeneity of variances was tested by Levene's test before ANOVA. Tukey's honestly significant difference test was used when the variances were homogeneous. Otherwise Tamhane's T2 test was used. Heatmaps and cluster analyses were performed using R software version 3.3.1 with pheatmap package (*R Core Team, 2013*; *Kolde, 2015*).

## RESULTS

### Total bacterial quantity and strain EDL933 measurement by qPCR

The background total bacterial quantities (the copy numbers of 16S rRNA gene) were $3.53 \times 10^9 \pm 1.23 \times 10^8$ and $5.82 \times 10^9 \pm 3.29 \times 10^8$ copies (g dry wt soil)$^{-1}$, respectively, for V3 and V4 regions. The total bacterial quantities in Treatments from E9 to E4 ranged from $7.00 \times 10^{10}$ to $3.53 \times 10^9$ and from $7.23 \times 10^{10}$ to $5.82 \times 10^9$ copies (g dry wt soil)$^{-1}$ for V3 and V4 regions, respectively. The total bacterial quantities of V3 and V4 regions in Treatments E9 and E8 were significantly ($p < 0.05$) higher than those in other treatments (Fig. S1). Compared with the background total bacterial quantity ($\sim 10^9$ copies (g dry

**Table 1  Detection and calculation of the added internal reference strain EDL933 (copies (g dry wt soil)$^{-1}$) in soil by qPCR (*fliC* gene and 7 × *fliC* gene) and iHAAQ method (Esch-V3 and Esch-V4).** Treatments E9 to E4 are corresponding to with 6 different concentrations of $10^9$ to $10^4$ CFU (g dry wt soil)$^{-1}$. Different letters indicate significant difference ($p < 0.05$) among 7 × *fliC* gene, Esch-V3 and Esch-V4 of each treatment detected by ANOVA test.

| Treatment | *fliC* | 7 × *fliC* | Esch-V3 | Esch-V4 |
|---|---|---|---|---|
| E9 | $4.79 \times 10^9 \pm 1.30 \times 10^9$ | $3.35 \times 10^{10} \pm 9.11 \times 10^8$ a | $4.67 \times 10^{10} \pm 9.07 \times 10^8$ a | $4.82 \times 10^{10} \pm 1.09 \times 10^9$ a |
| E8 | $5.96 \times 10^8 \pm 1.20 \times 10^7$ | $4.17 \times 10^9 \pm 8.38 \times 10^7$ ab | $3.59 \times 10^9 \pm 2.20 \times 10^8$ b | $4.37 \times 10^9 \pm 3.53 \times 10^8$ a |
| E7 | $5.28 \times 10^7 \pm 1.40 \times 10^6$ | $3.70 \times 10^8 \pm 9.80 \times 10^6$ a | $2.94 \times 10^8 \pm 1.17 \times 10^7$ a | $3.39 \times 10^8 \pm 1.13 \times 10^7$ a |
| E6 | $5.70 \times 10^6 \pm 4.62 \times 10^5$ | $3.99 \times 10^7 \pm 3.23 \times 10^6$ a | $2.82 \times 10^7 \pm 3.86 \times 10^6$ b | $4.08 \times 10^7 \pm 2.52 \times 10^6$ a |
| E5 | $5.65 \times 10^5 \pm 5.19 \times 10^4$ | $3.95 \times 10^6 \pm 3.63 \times 10^5$ ab | $3.48 \times 10^6 \pm 2.83 \times 10^4$ b | $6.47 \times 10^6 \pm 2.07 \times 10^5$ a |
| E4 | $8.11 \times 10^4 \pm 3.15 \times 10^3$ | $5.68 \times 10^5 \pm 2.21 \times 10^4$ c | $1.47 \times 10^6 \pm 7.66 \times 10^4$ b | $3.14 \times 10^6 \pm 1.25 \times 10^5$ a |

wt soil)$^{-1}$), the added strain EDL933 in the order of $10^4$ to $10^7$ copies (g dry wt soil)$^{-1}$ (Table 1) only had a slight influence on the total bacterial quantities in Treatments E4 to E7 (Fig. S1).

A standard curve of the *fliC* gene in the range of $3.60 \times 10^1$ to $3.60 \times 10^9$ copies $\mu L^{-1}$ was constructed for qPCR. The $C_t$ value of the standard sample containing $3.60 \times 10^1$ copies $\mu L^{-1}$ was $32.62 \pm 0.76$, and its standard deviation (SD) was greater than 0.5. However, the $C_t$ value of Cont ($33.05 \pm 0.98$) was even greater than the $C_t$ value of the standard sample containing $3.60 \times 10^1$ copies $\mu L^{-1}$. This qPCR result indicated that the detected gene in the control soil was not the *fliC* gene (Fig. S2).

In Treatments from E9 to E4, the $C_t$ values (from $13.73 \pm 0.05$ to $29.66 \pm 0.06$) of the *fliC* gene were all within the range of the standard curve, and all SD values were lower than 0.5. The *fliC* gene copies and added concentrations of strain EDL933 (data converted into $Log_{10}$) fitted well to a linear model ($R^2 = 0.999$) (Fig. S3). With or without intercept, the slopes were close to 1, indicating that the quantified *fliC* gene in the extracted soil DNA can be used to predict the concentrations of strain EDL993 in soil with high accuracy.

## Establishment and validation of the iHAAQ method

After quality filtering, there was a total of 465,220 high-quality sequences (66460 sequences per sample), and the sequences were classified into 35 phyla. At the genus level, strain EDL933 was identified and classified as genus *Escherichia-Shigella* in the SILVA database. The relative abundances of genus *Escherichia-Shigella* in Treatments from E4 to E9 were much higher than that in Cont, and they increased when more strain EDL933 was added to the soil (Fig. 2). The percentage of genus *Escherichia-Shigella* in phylum *Proteobacteria* was 89.76% in Treatment E9, compared with 0.07% in Cont.

After the *fliC* gene copies were multiplied by seven times (marked as 7× *fliC*), they were in the same order of magnitude of the absolute abundance of Esch-V3 and Esch-V4, except for Treatment E4 (Table 1). The ANOVA results showed that there was no significant difference ($p > 0.05$) between the 7× *fliC* and Esch-V4 in all the treatments, except for Treatment E4 (Table 1). In Treatment E4, both the *fliC* gene copy number and 7× *fliC* gene copy number were significant lower than the absolute abundances of Esch-V3 and Esch-V4.

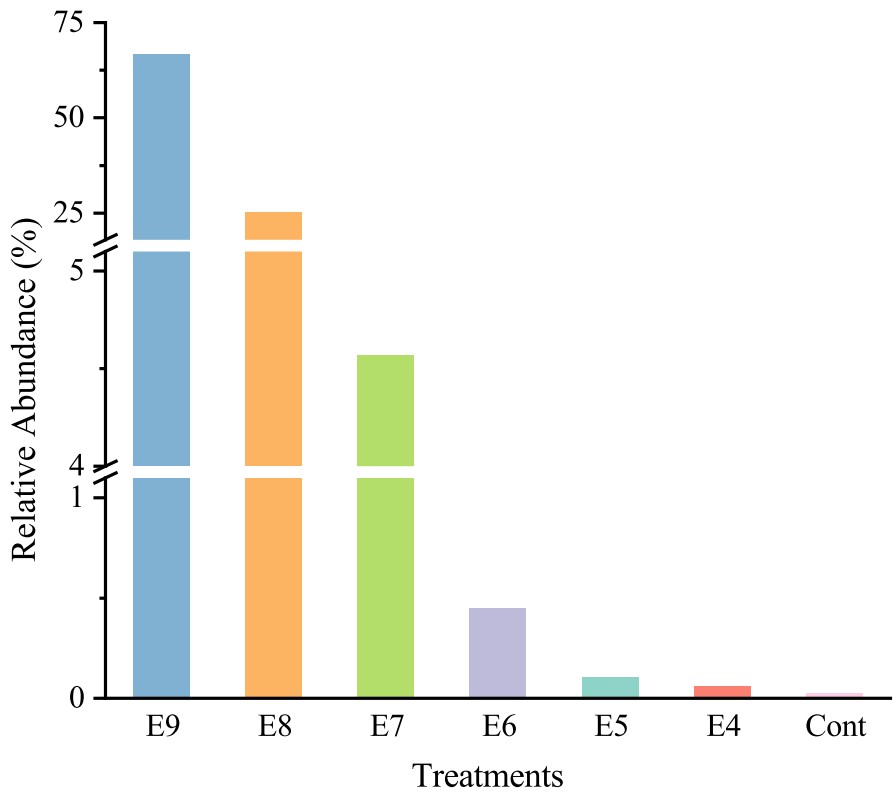

**Figure 2** **Relative abundances of *Escherichia-Shigella* genus in the different treatments.** The added internal standard strain EDL933 is classified as *Escherichia-Shigella* genus in the SILVA database. Treatments E9 to E4 are the same as described in Table 1, and Cont is the original control soil.

Furthermore, the $7\times$ *fliC* gene copies were linearly related to the calculated absolute copies of Esch-V3 and Esch-V4 (Table 1, Fig. 3). All the four linear equations fitted to the experimental data very well ($R^2$, 0.998–0.999), indicating that the $7\times$ *fliC* gene copies of the strain EDL933 can be reliably predicted from the absolute abundances of Esch-V3 and Esch-V4 (Fig. 3). Thus, it is apparent that the iHAAQ method based on the integration of qPCR and high-throughput sequencing can be used to calculate the absolute abundance (Figs. 1 and 4).

## Examining dynamics of soil bacterial communities by the iHAAQ method

From seven days to 28 days incubation, the residual PHE was declined rapidly from 22.01 to 8.44 µg (g dry wt soil)$^{-1}$ in Treatment P and from 7.53 to 4.38 µg (g dry wt soil)$^{-1}$ in Treatment W. Compared with Treatments P and W, Treatment S contained the highest residual PHE and had the slightest change during the 28-d incubation (47.79–60.45 µg (g dry wt soil)$^{-1}$). Following the procedures as shown in Fig. 1, the dynamics of soil bacterial community in Treatments P, W, and S were assessed by the iHAAQ method. The total bacterial quantities increased with the incubation time in Treatments P and W, but decreased with the incubation time in Treatment S (Fig. S4).

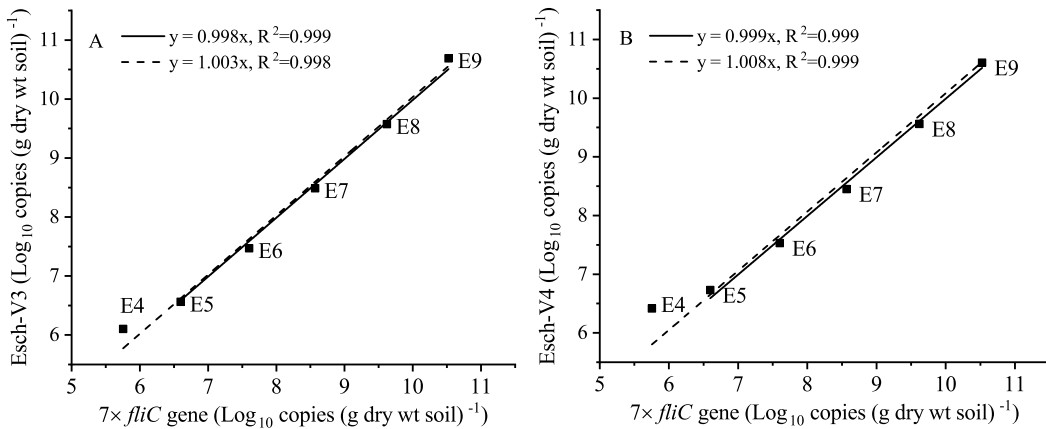

**Figure 3** **Linear regressions for the quantities of added internal standard strain EDL933 from qPCR and iHAAQ methods in the different treatments.** The $7 \times fliC$ gene copies of added internal standard strain EDL933 detected by qPCR. The Esch-V3 (A) and Esch-V4 (B) represent *Escherichia-Shigella* copies calculated by iHAAQ method with its relative abundance multiplying by the total bacterial 16S rRNA gene copies in V3 and V4 variable regions, respectively. All intercepts were fixed to the origin point (0). Treatments E9 to E4 are the same as described in Table 1. The solid and dash lines represent the linear models without and with Treatment E4, respectively.

There were mainly 17 phyla in which the relative abundances were greater than 0.1% in each of the samples (Table S1, Fig. S5). The dynamics of relative abundances reflected the soil bacterial composition changes during incubation (Fig. S5). Overall, the relative abundances decreased with the increase of incubation time in more than half of the phyla. The phyla of *Firmicutes*, *Proteobacteria*, *Planctomycetes*, *Bacteroidetes*, and *Actinobacteria* decreased by 5.58%, 1.93%, 1.59%, 1.28%, and 1.24%, respectively, in Treatment P; the phyla of *Proteobacteria*, *Firmicutes*, and *Verrucomicrobia* decreased by 9.24%, 3.41%, and 0.93%, respectively, in Treatments W; and the phyla of *Actinobacteria*, *Chloroflexi*, *Gemmatimonadetes*, and *Proteobacteria* decreased by 6.57%, 4.24%, 2.77%, and 1.40%, respectively, in Treatment S. However, the phylum *Firmicutes* in Treatment S increased substantially from 8.53% to 25.87%, and eventually it became the dominated bacteria in soil.

Nevertheless, relative abundance does not provide information about change of total bacterial quantities in soil. Based on the absolute abundances calculated by our proposed iHAAQ method, there were only five phyla showing decrease in both the relative and absolute abundances in Treatment P, while the relative and absolute abundances of other phyla showed opposite trend (Figs. 1, 5–7; Table S1, Fig. S5). As an example, the difference in relative and absolute abundances was especially evident for phylum *Proteobacteria* in Treatment P, in which the relative abundance decreased by 6.67%, while the absolute abundances increased by 42.22% and 21.02% in V3 and V4 regions, respectively. Such inconsistent result was also observed in Treatments W and S. In Treatment W, the relative abundance of four and six phyla decreased, while the absolute abundances of all phyla increased. In Treatment S, none of the absolute abundances of phylum increased, but the relative abundance of phylum *Firmicutes* increased by 17.54%.

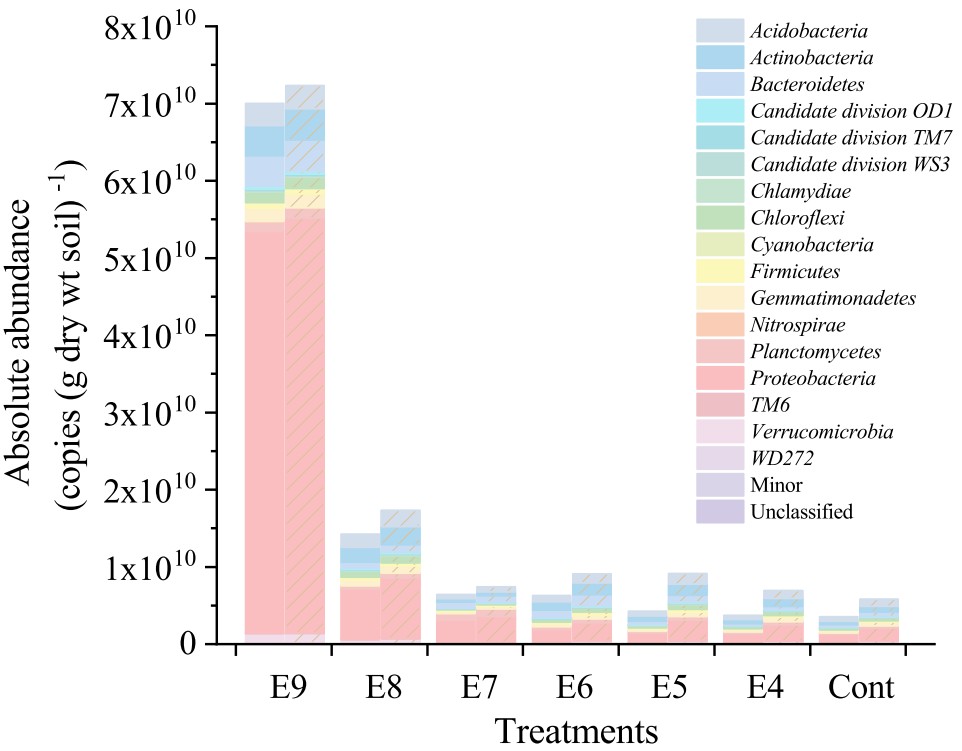

**Figure 4  Absolute abundances of the major phyla in different treatments.** All classified phyla with a relative abundance of <0.1% in a sample were combined and reported as Minor. Treatments E9 to E4 are the same as in Table 1, and the Cont is the original control soil. No shade and shadow histograms represent the phyla number quantified by the V3 and V4 regions of 16S rRNA gene, respectively.

Discrepancy between relative and absolute abundances also occurred at the genus level (Fig. 7, Table S1). In Treatment P, 135 and 57 opposite results in relative and absolute abundances were observed in V3 and V4 regions, respectively. While in Treatment W, 77 and 160 opposite relative and absolute abundances were found in V3 and V4 regions, respectively. Even more discrepancies (258 and 245 genera in V3 and V4 regions, respectively) were observed in Treatment S. While it was a general trend that the relative abundances of genera decreased while their absolute abundances increased in Treatments P and W (Table S1). The opposite trend was observed in Treatment S, in which the relative abundance increased while the absolute abundance decreased. For instance, the relative abundance of the dominant genus *Bacillus*, which belonged to the phylum *Firmicutes*, increased from 3.67% to 13.11% in Treatment S (Fig. 7). However, its absolute abundance decreased by more than 20% (from $8.88 \times 10^7$ to $5.26 \times 10^7$ copies (g dry wt soil)$^{-1}$ in V3 region, and from $1.10 \times 10^8$ to $8.69 \times 10^7$ copies (g dry wt soil)$^{-1}$ in V4 region).

The added strain WG5 in the W0 sample was also detected by high-throughput sequencing and classified as genus *Massilia*. The relative abundance of *Massilia* was higher than those in the P0 and S0 samples (<0.10%). The change in relative and absolute abundances of *Massilia* were also different in Treatment W after the soil was incubated at 28 °C for 28 days. Its relative abundances in the samples were 10.71% and 2.71%,

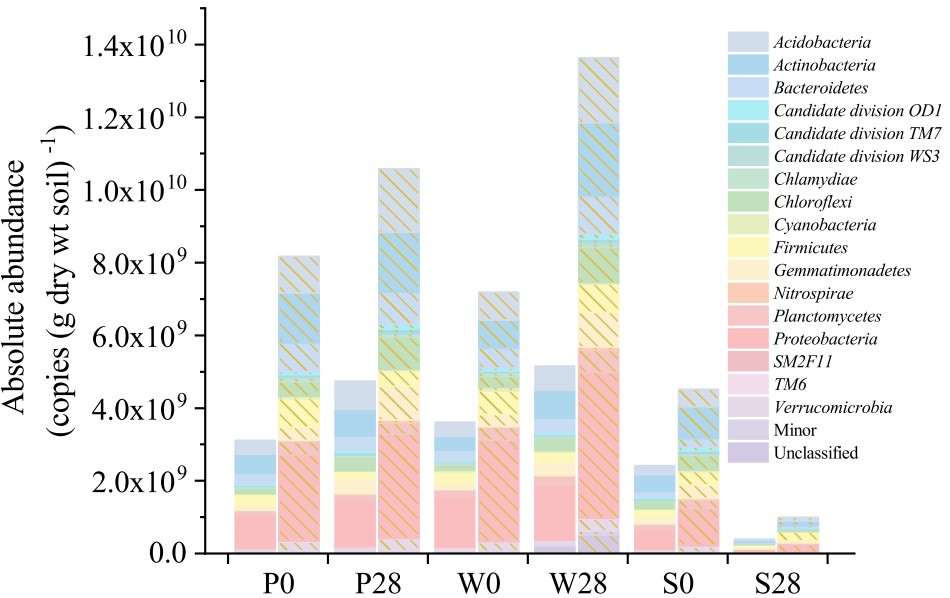

**Figure 5 Absolute abundances of the major phyla determined by the iHAAQ method in different soil samples.** All classified phyla with a relative abundance of <0.1% in a sample were combined and reported as Minor. P, S, and W represent the treatments of soil added with PHE (100 μg (g dry wt soil)$^{-1}$), NaN$_3$ (0.1%, w/w), and PHE (100 μg (g dry wt soil)$^{-1}$) and strain WG5 (approximately $1.00 \times 10^7$ CFU (g dry wt soil)$^{-1}$), respectively. 0 and 28 represent sampling at 0 and 28 days of incubation. No shade and shadow histograms represent the phyla number quantified by the V3 and V4 regions of 16S rRNA gene, respectively.

respectively, in Day 0 (W0) and Day 28 (W28) (Fig. 7), dropped almost three quarters (74.70%). Whereas its absolute abundances in W0 and W28 samples decreased from 3.86 $\times 10^8$ to $1.40 \times 10^8$ copies (g dry wt soil)$^{-1}$, and from $5.13 \times 10^8$ to $2.47 \times 10^8$ copies (g dry wt soil)$^{-1}$, representing less than two thirds (63.69%) and one half (51.84%) of decrease, respectively, in V3 and V4 regions.

Cluster analysis also showed the discrepancy between the relative and absolute abundances of bacteria in the treatments (Fig. 6, Fig. S6). Based on the relative abundances, the samples of P0, W0 and P28, S0 were clustered into two close groups, respectively, while the samples of W28 and S28 were far away from those groups (Fig. 6A). However, the cluster based on the absolute abundances showed totally different results. The samples of P28, W28 in V3 and V4 regions were clustered into the same group (Fig. 6B, Fig. S6). Meanwhile, the samples of P0, W0 and S0, S28 in V4 region were clustered into two different groups (Fig. 6B), which is more rational than the result in V3 region (Fig. S6).

## DISCUSSION

The V3 and V4 regions of 16S rRNA gene were universally used to detect the total bacterial quantity in soil via qPCR (*Reddy et al., 2012*; *Guinane et al., 2013*). *Youssef et al. (2009)* observed that there exists difference in soil bacterial community in V3 and V4 regions. Except for Treatment E9, all other treatments or samples showed that the total bacteria

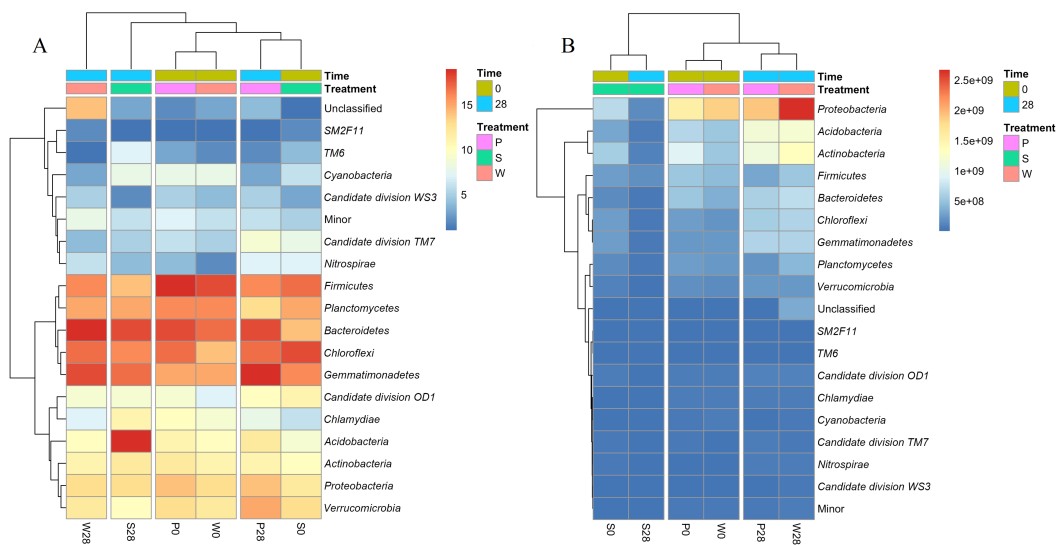

**Figure 6** **Heatmaps showing the relative abundances (A) and absolute abundances (B) determined by the iHAAQ method of the major phyla quantified by the V4 region of 16S rRNA gene.** All classified phyla with a relative abundance of <0.1% in a sample were combined and reported as Minor. P0, P28, S0, S28, W0 and W28 are the same as in Fig. 5. The color codes indicate the relative (A) or absolute (B) abundances, ranging from blue (low abundance) to red (high abundance).

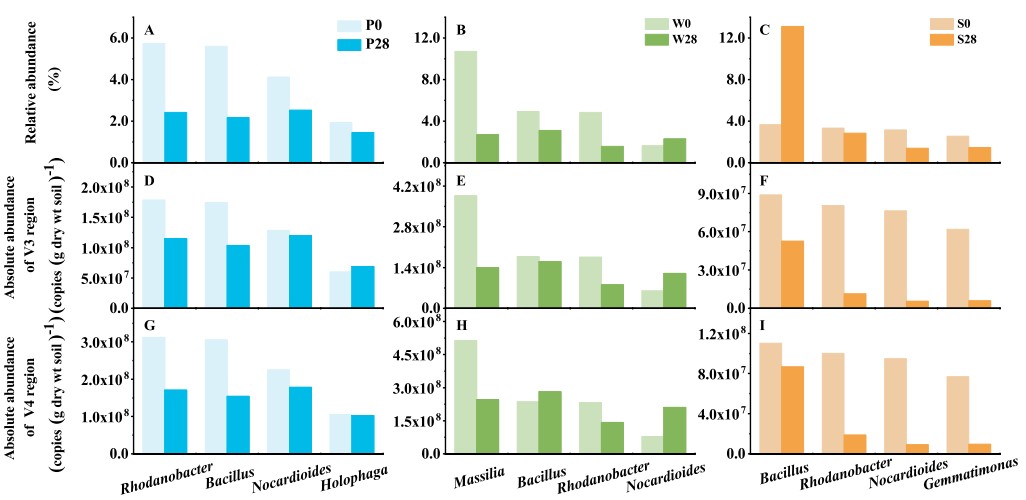

**Figure 7** **The representative genera of Treatments P, W, and S.** P0, P28, S0, S28, W0 and W28 are the same as in Fig. 5.

quantities in V4 region were significantly higher than those in V3 region. Comparing with the nearly complete 16S rRNA gene, the results of V3 region generally underestimated the total quantity of soil bacterial community, while the results of V4 region were usually comparable to it (*Youssef et al., 2009*). Meanwhile, *Wang et al. (2007)* also reported that V4 region has the highest reliability in classification among the 16S rRNA variable regions.

Thus, it was suggested that V4 region was the best choice of 16S rRNA gene variable regions for bacterial community characterization. Cluster analysis also proved that the results from the same region (V4) of qPCR and high-throughput sequencing were better than those from the different regions of them. Therefore, in the following discussion, we focused our discussion only on the results of 16S rRNA gene in V4 region.

Comparing with the selected taxa reported by *Dannemiller et al. (2014)*, it is very difficult to select a strain from soil to verify the iHAAQ method. As the previous research reported, the *fliC* gene of strain EDL933 can be precisely measured by qPCR (*Jun et al., 2011*; *Li & Chen, 2012*). Therefore, to test the accuracy of absolute abundance calculated by our method, an internal reference strain EDL933 was selected and introduced into the soil for validation. The high $R^2$ values of the linear relationships indicate strong association between the *fliC* gene and the actual added strain. The similar result was also found by *Tourlousse et al. (2017)* that the slopes of the spike-in read counts and input amounts were close to 1. Thus, we concluded the iHAAQ method based on qPCR and high-throughput sequencing in the same V4 region can be used to reliably predict the absolute abundance of *Escherichia-Shigella*.

The validation of the iHAAQ method was further verified by the added *Massilia* sp. strain WG5 in W0 sample. The absolute abundance of genus *Massilia* calculated by the iHAAQ method ($5.13 \times 10^8$ copies (g dry wt soil)$^{-1}$) was one order of magnitude higher than the added strain WG5 concentration ($1.00 \times 10^7$ CFU (g dry wt soil)$^{-1}$). After considering that one chromosome of strain WG5 has 9 copies of 16S rRNA gene (*Lou et al., 2016*), the calculated absolute abundance of *Massilia* was close to the added concentration of strain WG5.

When the iHAAQ method is applied to quantify soil bacterial community dynamics, both similar and opposite trends of relative and absolute abundances were observed, which agreed with early investigations (*Dannemiller et al., 2014*; *Prest et al., 2014*; *Smets et al., 2016*; *Stokell, Hamp & Steck, 2016*; *Zhang et al., 2017*). We observed that both the relative and absolute abundances of genus *Massilia* decreased in Treatments W after 28 days of incubation. The low residual PHE and its slow decrease in Treatment W indicated its low bioavailability, which might lead to the slight decrease of the genus *Massilia*.

In Treatment S, the addition of NaN$_3$ significantly decreased the absolute abundances of soil bacterial taxa, which agreed with the report of *Rouwane et al. (2016)*. The genus *Bacillus* became the dominant composition with the increased relative abundance but the decreased absolute abundance in the bacteria community of Treatment S. *Klein et al. (1994)* stated that some species in genus *Bacillus* were strongly resistant to NaN$_3$. It might result in the less decrease of genus *Bacillus* than other bacteria in Treatment S.

The relative abundance can reflect the composition changes of the microbial communities, while the absolute abundance can uncover the dynamics of the total quantities (*Reddy et al., 2012*; *Guinane et al., 2013*; *Dannemiller et al., 2014*; *Prest et al., 2014*; *Props et al., 2017*; *Zhang et al., 2017*). The relative abundance from high-throughput sequencing can only partially characterized the bacterial community (*Smets et al., 2016*; *Props et al., 2017*; *Tourlousse et al., 2017*). The actual microbial community dynamics can

be more objectively described by both the relative and absolute abundances from the iHAAQ method than by the relative abundance alone.

Furthermore, even if the relative abundance of samples had been obtained from the high-throughput sequencing, the absolute abundance of each taxon in the samples will be achieved with the supplementary qPCR analysis. It was recommended to use the same variable region of 16S rRNA gene for qPCR and high-throughput sequencing. Hence, the iHAAQ method is suitable to determine the absolute abundance of bacterial community and can be applied to studying the quantitative bacterial ecology in soil.

It is worthy to mention that the range of the valid absolute abundance of each taxon is limited by the high-throughput sequencing technique. Based on pyrosequencing data of a fungal aerosol population, the exact measure of absolute abundance of the fungal species can be achieved at the sufficiently high relative abundance (above $4.4 \times 10^{-4}$) (*Dannemiller et al., 2014*). In our study, the least sequence number of indigenous bacteria in soil were five reads (i.e., the least relative abundance was about $8.0 \times 10^{-5}$). The qPCR results showed that the total bacteria quantities in our samples were from $10^8$ to $10^{10}$ copies (g dry wt soil)$^{-1}$. Therefore, the indigenous bacteria in the soil samples can be detected by the iHAAQ method when their absolute abundances were above $10^3$ copies (g dry wt soil)$^{-1}$. With the increase of high-throughput sequencing depth, more rare species with lower abundance can be detected (*Agogue et al., 2011*; *Polka et al., 2015*), which may help to better characterize microbial communities.

## CONCLUSION

Our results indicate that the proposed iHAAQ method allows to obtain the absolute abundance of each taxon in soil bacteria communities by combining qPCR (total bacterial quantity) and high-throughput sequencing (relative abundance of each taxon), which could not be achieved separately by either method. The absolute abundance reflects the dynamics of a bacterial community more accurately than the relative abundance data. In our study, the iHAAQ method was successfully applied to characterize the soil bacterial communities with both relative and absolute abundances, which could provide more information than with the relative abundance alone by high-throughput sequencing. The proposed iHAAQ method could help to study soil quantitative bacterial ecology in future application.

### Funding
This work was supported by the National Natural Science Foundation of China (41471254, 41721001). The funders had no role in study design, data collection and analysis, decision to publish, or preparation of the manuscript.

### Grant Disclosures
The following grant information was disclosed by the authors:
National Natural Science Foundation of China: 41471254, 41721001.

## Competing Interests

The authors declare there are no competing interests.

## Author Contributions

- Jun Lou conceived and designed the experiments, performed the experiments, analyzed the data, prepared figures and/or tables, authored or reviewed drafts of the paper, approved the final draft.
- Li Yang conceived and designed the experiments, performed the experiments, analyzed the data, authored or reviewed drafts of the paper, approved the final draft.
- Haizhen Wang conceived and designed the experiments, contributed reagents/materials/analysis tools, authored or reviewed drafts of the paper, approved the final draft.
- Laosheng Wu conceived and designed the experiments, prepared figures and/or tables, authored or reviewed drafts of the paper, approved the final draft.
- Jianming Xu conceived and designed the experiments, authored or reviewed drafts of the paper, approved the final draft.

## Data Availability

The raw data obtained in this research are accessible via NCBI SRA (Sequence Read Archive; http://www.ncbi.nlm.nih.gov/sra/) under accession numbers SRP097773 and SRP105351.

## Supplemental Information

Supplemental information for this article can be found online at http://dx.doi.org/10.7717/peerj.4514#supplemental-information.

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
