# Peer review of "Assessing soil bacterial community and dynamics by integrated high-throughput absolute abundance quantification"

_PeerJ, doi:10.7717/peerj.4514_

## Round 0.1 · original submission · Major Revisions

· Academic Editor

Major Revisions

This manuscript employed an integrated method for quantification of microbial community in a mixed culture, which provides an alternative approach for a better understanding of the dynamics of absolute abundance of microbial community during phenanthrene biodegradation in soil. Both reviewers pointed out that the wording and gramma of this manuscript need to be improved greatly for a better interpretation of the results. In addition, the statistical analyses about the data should be addressed and presented clearly. A major revision is required and the authors need to provide a point-to-point response to the comments in details, before consideration of acceptance for publication.

·

Basic reporting

Lou et al have found that relative abundance and absolute abundance show opposite trend, and they proposed the iHAAQ method to quantitatively measure the absolute abundance. While I acknowledge the importance of combing the qPCR and high-throughput sequencing, some issues such as experimental design, figures and statistical analyses should be assess clearly. Please see the comments below. In addition, there are some small grammatical problems throughout the manuscript, especially in the results. There are also some long sentences that are hard to understand.

Experimental design

The molecular methods are well presented. However, no information about the number of replicates in each treatment was given in the methods. The power and significance of statistical tests are hard to asses if the replicates are missing. In addition, some analyses were not demonstrated in the methods, such as the cluster analysis.

Validity of the findings

The findings are meaningful and the conclusions are well stated. However, a lot of results about comparisons on relative and absolute abundance should be assessed in a statistical way. Any statistical comparisons on these values? Especially in Fig. 4, Fig. 5 and Fig. 7.

Additional comments

There are a lot of sentences in the results are actually the description of methods, such as how to calculate the absolute abundance. I think this type of information should be moved to the methods section.

In the discussion, a lot of results are repetitive. Please focus on discussing what you found and avoid repeating presenting your results.

Line34-35: It is for sure that, any advantage for this study?
Line47-48: The question about “who are they” is not assessed in community ecology. Community ecology is about the abundance and distribution of species, community assembly, species interactions, etc. In addition, you seems like talking about microbial species richness here, but you are assessing the abundance of microbes in this study.
Line57: Development of specific target genes? Please try to rephrase this sentence.
Line61: Again, I don’t think “community ecology” is used appropriately here.
Line193: Sequenced with primers? Please try to rephrase this sentence
Line220: ANOVA was used with SPSS
Line245: Please cite the figures in an appropriate way.

Table1: Are these values are mean values in different treatments? How about the standard errors? In addition, I think the test is based on post-hoc multiple comparisons? Which type of multiple comparisons were you using?
Fig.1: I think the “introducing the internal reference strain (IRS)” should be placed in the first row of this figure.
Fig.2: Can you provide the error bar here?
Fig.4: “Minor” is missing in this figure, although you mentioned in the legend. The phyla names should be italic.
Fig.4, Fig.5 and Fig.7: A lot of results about which one is higher or which one decreases are presented in the text. But are these comparisons statically validated? It is more appropriate if you can provide the error bars and statistical comparisons on these values.

Reviewer 2 ·

Basic reporting

The present manuscript described an iHAAQ method, which combined the high-throughput seq with qPCR (ideally V4 region of 16S rDNA), to reveal the absolute abundance of microbial community. Although such combination has been noted previously (Props, R. ISME J, 2016; Stammler, F. Microbiome, 2016; Zhang Z. Scientific reports, 2017), the study has its own merits: Validation of the reliability and accuracy of iHAAQ by an internal reference strain EDL933 and a strain WG5 with PAH degrading ability. The study is well designed and conducted, and obtained data are sufficient to support the conclusions. Overall it is an interesting paper.

Experimental design

The study is well designed and conducted, and obtained data are sufficient to support the conclusions.

Validity of the findings

-It has been widely admitted that template-specific Illumina sequencing artifacts may lead to biases in the perceived abundance of certain taxa, that is, the wide range of experimental variables may introduce bias at all steps of a typical 16S-seq workflow. Did authors take any measure to minimize the bias effects?
-The discussion needs to include some comments about synthetic spike-in standards (Nucleic Acids Research, 2017, Vol. 45, No. 4 e23 doi: 10.1093/nar/gkw984)
- 16S rRNA is variable in copy numbers in bacterial genomes, more specific information should be given in Line 210-212.
-Lots of phrases not sufficiently concise, for example, the redundancy in Line 69-74, is it really necessary to apply detailed data for illustration?
-The information in line 94-97 was contradictory to that in line 90-92.
-Line 126, PHE, full name should be given on its first appearance.
-Readable, but does need some grammer and misc corrections throughout.

---

## Round 0.2 · accepted · Accept

· Academic Editor

Accept

The authors have addressed all the comments from the reviewers and myself.